# Fault Diagnosis Methods for an Artillery Loading System Driving Motor in Complex Noisy Environments

**DOI:** 10.3390/s24030847

**Published:** 2024-01-28

**Authors:** Wenkuan Huang, Yong Li, Jinsong Tang, Linfang Qian

**Affiliations:** 1School of Mechanical Engineering, Nanjing University of Science and Technology, Nanjing 210094, China; hwk333@njust.edu.cn (W.H.); yongli@njust.edu.cn (Y.L.); tangjs@njust.edu.cn (J.T.); 2Northwest Institute of Mechanical and Electrical Engineering, Xianyang 712099, China

**Keywords:** fault diagnosis, artillery driving motor, attention mechanism, AdaBoost, noise

## Abstract

With the development of modern military technology, electrical drive technology has become a power source for modern artillery. In fault monitoring of a driving motor mounted on a piece of artillery, various sensors are susceptible to interference from the complex environment, both inside and outside the artillery itself. In this study, we creatively propose a fault diagnosis model based on an attention mechanism, the AdaBoost method and a wavelet noise reduction network to address the difficulty in obtaining high-quality motor signals in complex noisy interference environments. First, multiple fusion wavelet basis, soft thresholding, and index soft filter optimization were used to train multiple wavelet noise reduction networks that could recover sample signals under different noise conditions. Second, a convolutional neural network (CNN) classification module was added to construct end-to-end classification models that could correctly identify faults. The above basis classification models were then integrated into the AdaBoost method with an improved attention mechanism to develop a fault diagnosis model suitable for complex noisy environments. Finally, two experiments were conducted to validate the proposed method. Under motor signals with varying signal-to-noise ratios (SNRs) noises, the proposed method achieved an average accuracy of 92%, surpassing the conventional method by over 8.5%.

## 1. Introduction

With the development of modern artillery towards automation and intelligence, the motor system has gradually evolved to be the power system for such artillery due to its highly efficient output, fast response, and excellent reliability [1]. However, because modern artillery now involves a high degree of automation, any failure of the electrical system can greatly affect the effectiveness of the artillery in combat, and in severe cases can lead to disastrous consequences [2]. As a result, accurate fault diagnosis is required to detect anomalies in the motor system in a timely manner, thus ensuring the safety of equipment and personnel, as well as improving the efficiency of logistics support, which is extremely important.

Figure 1 shows a shock vibration test near the trunnion of a self-propelled piece of artillery that uses a motor to drive the coordinating arm for drug delivery. Experiments have found that the vibrations inside the artillery were so intense that the vibration signals collected by the sensors were severely disturbed [3,4]. The working environment of artillery is characterized by harsh conditions including strong background noise and sensor impact caused by firing. Moreover, artillery falls under typical non-static working conditions. These unstable excitations and complex noisy conditions directly contribute to challenges in extracting fault characteristic signals from the driving motor of artillery.

Common motor faults are usually detected using vibration, magnetic, and current signals [5,6,7]. Vibration signal detection has obvious fault characteristics and possesses a high level of detection accuracy for mechanical faults, e.g., motor bearing faults [8,9,10]. Magnetic signals are sensitive to flux changes caused by any magnetic field imbalance and have recently become an important area of research [11,12]. Current signals have the advantages of being easy to collect and monitor and very precise, with low noise [13,14,15,16]. Strong recoil during firing can cause transient eccentric vibrations of the motor rotor, which can also cause significant interference with magnetic signal-based diagnostic methods. Current signals are relatively less affected by shock vibrations, and the anti-jamming cable also has a certain shielding effect against various electromagnetic countermeasures on the battlefield. Considering these advantages and disadvantages, the current signal-based fault diagnosis method is still chosen for artillery motor fault diagnosis.

Different current signal data processing techniques and intelligent diagnosis algorithms have been widely used for motor fault diagnosis tasks. Huang et al. [17] used an autoencoder and a recurrent neural network (RNN) to detect motor faults through current signals. Widodo et al. [18] proposed a fault diagnosis method based on principal component analysis (PCA) and a support vector machine (SVM), which is simple and effective for extracting motor fault features. These signal processing methods have developed rapidly in the last decade, with solid theoretical foundations and good interpretability. At the same time, they address current problems such as signal denoising and weak feature extraction. However, this approach usually requires expert experience in fault characteristics.

In recent years, motor fault diagnosis based on deep learning (DL) has attracted widespread attention as an end-to-end approach that does not require expert experience or characterization [19,20,21,22,23]. Considerable literature on motor fault diagnosis has emerged around the topic of deep learning. Suawa et al. [24] proposed a fusion method using data-level sensors and deep learning, proposing Convolutional Long Short-Term Memory (CNN-LSTM), which is a combination of two deep learning methods in order to diagnose Brushless Direct Current motor faults. Husari et al. [25] proposed a hybrid architecture, namely a 1D convolutional neural network-long short-term memory (1DCNN-LSTM) and a 1DCNN-gated recurrent unit (GRU)-based approach, for early inter-turn fault diagnosis. Li et al. [26] proposed a wavelet kernel net, which used a wavelet basis instead of convolutional kernels, and combined the advantages of deep learning and classical signal processing methods. Zhao et al. [27] designed a filter denoising network and trained the filter threshold using the deep residual shrinkage networks network. In light of the exceptional anti-noise capabilities offered by fusion wavelet and deep learning methodologies, this study employs these techniques to construct networks for noise reduction and feature extraction. However, existing research predominantly focuses on analyzing motors in relatively stable operating environments, where even if the test signal is disrupted by noise, its signal still maintains a comparatively stable signal-to-noise ratio (SNR) and noise distribution.

In order to solve the generalization ability of fault diagnosis models to complex noise, this study draws on the advantages of ensemble learning and attention mechanisms. Ensemble learning performs the diagnostic task by iteratively constructing and combining multiple classifiers, which greatly improves the generalization ability of the model. Long et al. [28,29] used the AdaBoost ensemble learning algorithm to improve the diagnostic accuracy of unbalanced experimental datasets; this proved a significant inspiration for fault diagnosis in artillery motors. The idea of giving extra weight to specific test samples with extreme values within a distribution is consistent with the attention mechanism. The attention mechanism comes from the study of human vision: it works by selecting the most relevant information from amongst a given set and concentrating it, thereby improving the recognition and classification performance of an intelligent algorithm. The attention mechanism has been shown to play a clear role in a wide range of studies, but its applications are mainly in the fields of vision, image classification, and natural language translation, while its application in the field of fault diagnosis is still under development [30,31,32,33].

The main contributions of this paper are as follows:We designed a fusion wavelet kernel composed of multiple wavelet basis and applied it to fault diagnosis in artillery motors. This approach can efficiently process the current signals of the artillery motors under different operating conditions and support subsequent fault diagnosis.We propose a filter optimization method combining indexing and soft thresholding to filter the fused wavelet basis and achieve noise reduction for signals with different noise.We propose a modified version of the AdaBoost ensemble learning algorithm based on the attention mechanism. In addition to the weights assigned by the original AdaBoost algorithm, more attention is allocated to the optimal base classifier under the current conditions through similarity analysis of filtered samples. The final diagnosis model constructed according to this method is named the Denoising Fault Diagnosis Attention Network (DFAN). This approach further reduced the interference of raw sample noise on fault diagnosis and enabled the final classifier to be very generalizable and robust.

The rest of the paper is organized as follows: Section 2 presents the relevant preparatory work and the theoretical basis. Section 3 describes the implementation of the method in detail. Section 4 is devoted to a discussion of the experiments. Section 5 is devoted to the conclusion.

## 2. Preparatory Work

### 2.1. Current Signal for Diagnosis of a Fault in the Artillery Driving Motor

A typical motor-driven artillery automatic loading system is shown in Figure 2. It is a typical mechatronic system, which consists of an ammunition swing arm, an ammunition delivery mechanism, and a motor drive system to provide power. When the artillery is loaded, the arm is driven by an electric motor to rotate around the shaft, and the ammunition delivery mechanism is driven to complete the spatial conversion from one magazine to the next. At the same time, the attitude adjustment motor drives the bullet feeding mechanism to rotate around the rotating axis to realize the attitude adjustment from the bullet receiving position (coaxial with the exit of the shell warehouse) to the ammunition transporting position (coaxial with the artillery barrel). The process includes two steps of work:(a)The ammunition delivery mechanism resets from the position of ammunition delivery (barrel) to the position of receiving ammunition (magazine).(b)The ammunition delivery mechanism carries ammunition from the position of receiving ammunition (magazine) to the position of ammunition delivery (barrel).

A governing equation for the driving motor of the artillery loading mechanism was constructed.

The voltage element can be expressed as [34]:(1)$$\left\{\begin{array}{c} u_{d}=\frac{d\lambda_{d}}{dt}-\lambda_{q}\omega_{r}+R_{s}i_{d}\\ u_{q}=\frac{d\lambda_{q}}{dt}-\lambda_{d}\omega_{r}+R_{s}i_{q} \end{array}\right.$$

Here, *u_d_* and *u_q_* are the stator voltage d and q-axis components, *i_d_* and *i_q_* are the stator current *d* and *q*-axis components, $\lambda_{d}$ and $\lambda_{q}$ are the stator flux *d* and *q*-axis components, *R_s_* is the stator resistance and $\omega_{r}$ is the rotor electrical angular velocity.

The kinetic element can be formulated as [34]:(2)$$T_{e}-T_{l}=J\frac{d\omega_{m}}{dt}+B\omega_{m}$$

Here, *T_e_* is the motor actuation torque, *T*_l_ is the load torque, *J* is the moment of inertia, *B* is the friction coefficient, $\omega_{m}$ is the mechanical angular velocity of the rotor, $\omega_{r}=p\omega_{m}$ and *p* is the number of rotor poles in the motor.

During motor control, given the planning curve, the position loop outputs the command speed; the loop outputs the instruction current; and the current loop outputs the command voltage. Simultaneously, Equations (1) and (2) show that:(3)$$\begin{array}{l} I=f\left(T,J,B,\lambda ,\omega \right)\\ I=\left[\begin{array}{c} i_{d}\\ i_{q} \end{array}\right] \end{array}$$

The artillery driving motor torque *T* and rotor angular velocity ω caused by changes in the operation of the type of fault, the fault mechanism of the change of the moment of inertia *J*, the wear failure of the change of the friction coefficient *B* and the motor internal fault of the change of the stator flux linkage λ respond to the current ***I***. Three of the motor currents $I_{a},I_{b},I_{c}$ have a simple projection relationship with ***I*** and can be obtained directly from the motor monitoring equipment. In order to perform fault diagnosis on an artillery driving motor via a three-phase current, the core task is to extract features of each fault type from the current.

### 2.2. Signal Preprocessing

#### 2.2.1. Wavelet Transform

As a signal analysis tool that can obtain joint information in the time and frequency domain, the wavelet transform has been widely used to deal with non-stationary signals.

Suppose that a current signal including noise is collected from the motor; this can be expressed as follows:(4)x(t)=s(t)+n(t)

Here, x(t) is the measurement signal; s(t) is the characteristic signal; and n(t) is the signal containing multiple noisy interference signals.

The convolution of the function x(t) with a suitable wavelet basis function yields a decomposition in frequency and time:(5)$$W_{a,b}\left(x(t),\psi (t)\right)=\frac{1}{\sqrt{a}}\int_{-\infty }^{+\infty }x\left(t\right)\psi^{*}\left(\frac{t-b}{a}\right)dt$$

Here, $W_{a,b}(\cdot )$ denotes the wavelet coefficients obtained from the decomposition, *a* is the scale parameter proportional to the inverse of the center frequency, *b* is the translation parameter of the localization signal, $\psi^{*}(\cdot )$ is the complex conjugate wavelet of a mother wavelet basis ψ(⋅) scaled by the displacement, and *t* is the time step.

Adjusting the scale parameter can change the frequency of the wavelet, thus affecting the resolution in both the time and frequency domains and then achieving the purpose of displaying different details of the same signal. In past studies, different wavelet basis have been designed to reveal hidden features of non-stationary signals. Wavelet transforms composed using different scales and wavelet basis can be thought of as filters with different characteristics and frequency bands.

The first important aspect to improve the fault diagnosis effect is, therefore, the selection of an appropriate scale parameter and wavelet basis for the current signal of the artillery driving motor.

#### 2.2.2. Wavelet Signal Denoising

Wavelet transform is an important application of signal denoising. Electronic devices used in modern equipment such as radar, wireless communications, satellite navigation, and electronic countermeasure systems, among others, are very complex. In the actual battlefield environment, artillery signals are inevitably mixed with a large amount of noise due to self-interference and mutual interference of electronic devices.

After wavelet decomposition of the original signal, features tend to focus on large wavelet coefficients. The wavelet coefficients of the noise are uniformly distributed and small. By choosing an appropriate threshold, only signals consisting of wavelet coefficients larger than the threshold are retained for denoising purposes. There are two main approaches to thresholding [12]. The hard threshold is discontinuous in the real number domain and changes from continuous to step-like at the threshold point; this can easily lead to oscillations in the signal reconstruction and produce the pseudo-Gibbs phenomenon. The soft threshold overcomes the drawback that the hard threshold function is discontinuous at the threshold, but it has a constant deviation from the original wavelet coefficients, which reduces the similarity between the denoised signal and the original signal. The threshold function is shown in Figure 3. The threshold function can be defined as follows:(6)$$F(W,\tau )=\left\{\begin{array}{c} \;0\;\;\;\;\;\;\;\;\;\;\;\;\;\;\;\;\;\;\;\;\;\;\;\;\;\;\;\;\;,\;\left|W\right|<\tau \\ \mathrm{sgn}\left(W\right)\left(\left|W\right|-a\tau \right),\;\left|W\right|⩾\tau \end{array}\right.$$

Here, τ is the threshold and W are the coefficients after thresholding. When *a* = 0, $F\left(\cdot \right)$ is a hard threshold function. When a=1, $F\left(\cdot \right)$ is a soft threshold function. When 0<a<1, $F\left(\cdot \right)$ is a compromise between soft and hard thresholds.

The second important aspect is the design of an appropriate threshold function for the wavelet coefficients to achieve good noise reduction.

### 2.3. Reinforcement Learning for Building a Fault Classification Model

The selection of the optimal wavelet basis and threshold for the wavelet coefficients is equivalent to the optimization problem for the filter parameters. Traditional wavelet filter optimization methods use a fixed, pre-given filter parameter, that is, a fixed band assignment. However, the sample noise of the artillery driving motor is mixed and uncertain. Correspondingly, the filter chosen based on the training set may not be optimal, since the optimal parameters for a particular test set may not be included.

In this paper, filter parameters with certain effects are not discarded directly, but the reinforcement learning method is used to attach appropriate weights to these filters so that they play a role in the final faulty classifier.

#### 2.3.1. The AdaBoost Algorithm

The AdaBoost [35] algorithm combines weak classifiers to obtain strong classifiers. The algorithm is implemented by varying the data distribution. The weight values are determined by the accuracy of each sample in the current training set and the accuracy of the last overall classification. The new dataset with modified weights is sent to the next classifier for training, and finally, the classifier obtained from each training is fused as the final decision classifier, a weighted iterative process, as shown in Figure 4. The shaded samples are those that were misclassified and whose weights required changing. The samples, represented by circles of different colors, should have been categorized separately.

The specific procedure of the algorithm is as follows:(1)Randomly select m sets of training data from the sample space and initialize the distribution weights of the test data:
(7)$$D_{1}(i)=\omega_{1}(i)=1/m,i=1,2,\cdots ,m$$

Several weak classifiers are trained through the initial data distribution and the best one is selected as the first base classifier, *h*_1_.

(2)Iterative training: t=1,2,⋯,T;
(a)The first *t* wheel base classifier *h_t_* on the distribution of ***D****_t_* error:
(8)$$e_{t}=\sum_{i=1}^{m}D_{t}\left(i\right)\left(\;h_{t}\left({\vec{x}}_{i}\right)\neq y_{i}\right)$$
(b)Take the partial derivative of the loss function $l_{exp}(\beta_{t}|D_{t})$ and find the zero solution to obtain the weight of *h_t_* in the final classifier:(9)$$\beta_{t}=\frac{1}{2}\ln\left(\frac{1-e_{t}}{e_{t}}\right)$$(c)To update the training sample weight distribution ***D****_t+_*_1_:(10)$$\begin{array}{l} D_{t+1}\left(i\right)=\frac{D_{t}\left(i\right)\exp\left(-\beta_{t}y_{i}h_{t}\left({\vec{x}}_{i}\right)\right)}{Z_{t}}\\ Z_{t}=2\sqrt{e_{t}\left(1-e_{t}\right)} \end{array}$$


Here, *Z_t_* is a normalization constant.

A base classifier with a small classification error rate has a large weight, and a base classifier with a large classification error rate has a small weight. We then obtain *T* as a linear combination of the base classifiers. Based on the linear combination, the eigenfunctions are transformed to obtain the results of the strong classifier.
(11)$$H_{\mathrm{final}}=sign\left(\sum_{t=1}^{T}\beta_{t}h_{t}\right)$$

The AdaBoost classifier can remove some unwanted features of the training data and focus on the important features. At the same time, it has the property that the upper bound on the classification error rate steadily decreases with the amount of training and no overfitting occurs.

#### 2.3.2. Attention Mechanism

An attention mechanism is a well-known concept in the field of NLP. It can be understood as selecting and focusing on a small amount of important information from a large amount of information while ignoring most of the unimportant information. This idea still holds in the field of fault diagnosis. The essence of an attention mechanism is weighted summation [36].

In terms of practical engineering practice, this study notes that artillery loading devices drive motor faults with diversity and specificity. That is, the health status of the actuation motor of each piece of artillery is different and, moreover, there is a mixture of different noise levels. In a specific working condition, the fault diagnosis model trained by the samples collected by a specific artillery driving motor cannot be directly applied to other working conditions or other artillery driving motor fault diagnosis problems. In mathematical language, this can be expressed as:(12)$$P_{s}(x_{s},y)\neq P(x,y)$$

Here, ***x*** denotes the data used for fault diagnosis, *y* is the corresponding classification label of the data, P(⋅) is the joint probability distribution of ***x*** and *y*, the subscript s denotes the situation for a particular artillery driving motor and the subscripts without it denote the global situation. The poor performance filter in the global sample space is likely to be optimal depending on the sample distribution of a particular piece of artillery.

In this paper, we improved the AdaBoost algorithm by introducing an attention mechanism. After training the weights of the base classifier model, the weights were again optimized so that the classification model performed better in fault diagnosis on artillery motors in complex noisy environments.

## 3. Proposed Approach: Adaptive Fusion of Various Wavelet Basis to Extract Each Feature in the Signal

### 3.1. A Filter Framework Based on Fused Wavelet Convolution Method

Based on the conclusions drawn in Section 2.2.1, the selection of appropriate wavelet scale parameters and wavelet basis is an important aspect of signal processing. The current signal of an artillery motor contains many types of fault features and many types of noise; these are difficult to manage on a single wavelet basis, and the wavelet scale parameters are difficult to learn using conventional methods. This paper proposes a fused wavelet convolution approach to address the problem.

The form of the wavelet transform is similar to the 1D convolution case. By replacing the output of the original convolutional layer with the output of the wavelet basis, we can train the parameters of the wavelet basis using the method of training 1D convolutional layers [26].

Equation (5) can be simplified into convolution form:(13)$$W_{a,b}=\psi (t)*x(t)$$

Here, $W_{a,b}$ is the wavelet coefficient that replaces the output of the convolutional layer, and ψ is the trainable wavelet basis.

An effective approach is to combine multiple wavelet bases into a new wavelet basis and then perform data-driven pruning on the fused wavelet basis. Suppose that there are *C*_0_ basic wavelet bases at the beginning and *C* fusion wavelet bases left after fusion and pruning:(14)$${\psi^{\prime }}_{i}=\sum_{j=1}^{C_{0}}p_{i.j}\psi_{j},i=1,2,\cdots ,C$$

Here, ${\psi^{\prime }}_{i}$ represents the new fusion wavelet basis and $\psi_{j}$ denotes the original wavelet base.

We need to keep track of the real-time importance of each filter as we train the model. The fusion weight $p_{i,j}$ can be defined as:(15)$$p_{i,j}=\mathrm{Softmax}(-D(\psi \prime_{i},\psi_{j})\times \varepsilon ),\;j=1,2,\cdots ,C_{0}$$

Here, D(⋅,⋅) denotes a distance measurement: as a rule of thumb, the Euclidean distance works best. The parameter ε is capable of being acquired through learning. The Softmax() function guarantees that $\sum_{j=1}^{C_{0}}\;p_{i,j}=1$.

The logic of this weighted design is that if a wavelet basis is significantly different from other wavelet bases, and this wavelet basis plays an active role in the training process, then it should be represented. At the same time, the maximum possible use of large difference wavelet bases is also conducive to improving the generalization of the model. We used KL divergence [37] to design an importance index for the convolutional wavelet kernel:(16)$$I_{i}=\frac{1}{C}\sum_{m=1}^{C}\sum_{n=1}^{C_{0}}\;p_{i,n}\log\frac{p_{i,n}}{p_{m.n}},\;i=1,2,\cdots ,C$$

Based on the importance index, *C* important bases were selected; we then generated Equation (17) to implement the fused wavelet convolution:(17)$$W_{c}=x(t)*{\psi_{c}}^{\prime }(t)$$

The subscript *c* denotes the *c*th output channel. The fused wavelet convolution procedure is shown in Figure 5.

### 3.2. Filter Optimization and Selection

As mentioned in Section 2.2, another key task for wavelet denoising is to determine the threshold of the wavelet coefficients at each scale.

In this paper, residual shrinkage networks were used to learn thresholds and fine-tune parameters. We named this the wavelet denoising layer (WDL); its network structure is shown in Figure 6:

The threshold function of Equation (6) is updated as follows:(18)$$F(W_{c},\tau )=\left\{\begin{array}{c} 0,-\tau -B<W_{c}<\tau -B\\ W_{c}+B-\tau ,W_{c}⩾\tau -B\\ W_{c}+B+\tau ,W_{c}⩽-\tau -B \end{array}\right.$$

*B* is the fine-tuning parameter of the threshold and τ is the threshold at which the network learns.

The wavelet denoising layer can be expressed as Equation (18). W and $\tilde{W}$ are the inputs and outputs of the wavelet denoising layer, respectively.
(19)$$\tilde{W}=F\left(W,\tau \right)+W$$

Next, a loss function was designed to optimize the filter. According to the literature, energy is a widely used metric for finding wavelet filters with defect-related features. The central idea is that the energy of the wavelet coefficients in the defect-dependent band is higher than in the other bands. The energy can be calculated from the corresponding wavelet coefficients. The design was based on energy loss:(20)$$l_{isf}=-\frac{L}{C}\sum_{i=1}^{C}\frac{\sum_{j=1}^{L}{\hat{W}}_{i}{\left(j\right)}^{4}}{{\left(\sum_{j=1}^{L}{\hat{W}}_{i}{\left(j\right)}^{2}\right)}^{2}}$$

Here, $\hat{W}$ is the envelope of $\tilde{W}$:(21)$$\hat{W}=\sqrt{{\tilde{W}}^{2}+\mathrm{Hilbert}{\left(\tilde{W}\right)}^{2}}$$

After optimization, there are C-wavelet filters corresponding to the output of the C-channel.

Finally, the optimal filter was selected from the optimized C-wavelet filter, defined as hard filter selection in this paper.

The literature proposes a soft filter selection that preserves all features and implicitly selects the optimal filter. Considering that the FWConv used in this paper has already performed the dynamic fusion of important wavelet basis, the computational load is greatly increased if multiple mixed wavelet basis are retained, so only the optimal filter was used, and the others were discarded. In addition, a more efficient attention mechanism was used as the weight allocation metric for each round of wavelet basis. The structure of the index-based soft filtering module is shown in Figure 7.

Since energy is widely used to construct band selection indices in wavelet transforms, we constructed a new energy-based index. Note that the N-wavelet basis in FWConv leads to N-energy features and the channels should be divided into N-groups and their relative indices computed. However, FWConv dynamically fuses important wavelet bases, making packet computation complex and resource-intensive. A simple idea is to use convolution to compute the local relative index. Based on channel energy, the index (soft selection) can be expressed as:(22)$$\tilde{\omega }=\mathrm{sigmod}(\mathrm{conv}(E))$$
(23)$${\tilde{x}}_{out}={\tilde{W}}_{c}⊙{\tilde{\omega }}_{c}+{\tilde{W}}_{c}\;,\;\mathrm{if}\;c=\mathrm{argmax}(\tilde{\omega }(i))$$

Finally, the module consisting of fused wavelet convolution, soft thresholding, and index-based soft filtering was named the Denoising Wavelet Convolutional Network (DWCN).

The synthesis of Equations (13), (19) and (23) can be obtained as follows:(24)$${\tilde{x}}_{out}=\mathrm{DWCN}(x)$$

Referring to the approach used in ref. [38], we constructed a simple CNN multi-class classification module. The general case consists of a 1D convolutional layer, a BN layer, ReLU, a max pooling layer, and FC layers.

For class labels $\tilde{y}$, there is cross-entropy loss as follows:(25)$$l_{cls}=-\sum_{i=1}^{N}y_{i}\log({\tilde{y}}_{i})$$

Here, $y_{i}$ is the real label of x. *N* is the number of categories of labels *y.*

In the proposed approach, the DWCN is connected to the CNN classification module, and an end-to-end fault classifier can be trained; this is named the base classifier. The *l* oss function expression of the base classifier is:(26)$$l_{base}=l_{cls}+kl_{isf}$$
where *k* is the tradeoff parameter and can be determined via hyperparameter search methods, such as a grid search [23].

### 3.3. Improved AdaBoost Multi-Classification Algorithm Based on an Attention Mechanism

In this paper, the AdaBoost algorithm used many base classifiers to integrate a final classifier. After iterating the T-wheel using the AdaBoost method described in Equations (7)–(11) in Section 2.3, *T* base classifiers were obtained and a strong classifier was integrated to resolve the fault-diagnosis problem in the artillery drive motor.

However, in practical engineering applications, a global sample space is used in the model training process. However, when sampling a certain artillery-driven motor for fault diagnosis, there are insufficient labeled samples for pretraining. Due to the different health status and working conditions of the motors, the SNR and noise form of the noise contained in the samples used are also different, and the distribution of the samples is also different from the global sample space. To improve the diagnostic effect, it is possible to redistribute the weights of the base classifier through the attention mechanism. The full algorithmic flowchart is shown in Figure 8.

For a set of motor samples ***I*** on a certain piece of artillery, the output processed by the DWCN module of base classifier *h_t_* is represented as $x_{t,out}$, and the healthy noise-free samples under corresponding working conditions are represented as s. In order to facilitate the direct similarity comparison between the two types of samples, PCA was used to reconstruct the two types of samples. PCA is a simple and efficient feature extraction method that preserves the most original information and minimizes the reconstruction error.

The $x_{t,out}$ and s were constructed as autocorrelation matrices of the same dimension and PCA was then performed to obtain the feature vectors $x_{t,PCA}$ and $s_{PCA}$ of the same dimension. A similarity analysis was then performed:(27)$$\begin{array}{l} c_{t}=\mathrm{sim}\left(x_{out},\;s\right)=\frac{{\bar{x}}_{t,\mathrm{PCA}}\cdot {\bar{s}}_{\mathrm{PCA}}}{\left|{\bar{x}}_{t,\mathrm{PCA}}\right|\times \left|{\bar{s}}_{\mathrm{PCA}}\right|}\\ \alpha_{t}=\mathrm{softmax}\left(c_{t}\right)=\frac{\exp\left(c_{t}\right)}{\sum_{t=1}^{T}\exp\left(c_{t}\right)} \end{array}$$

Here, ${\bar{x}}_{t,\mathrm{PCA}}$ and ${\bar{s}}_{PCA}$ are the mean values of the feature vectors obtained after PCA processing of the selected samples; $c_{t}$ is the similarity and $\alpha_{t}$ is the attention weight assigned to the classifier based on. By rewriting Equation (11), the classification label of the final classifier is described by Equation (28):(28)$$\begin{array}{l} H_{\mathrm{final}}=sign\left(\sum_{t=1}^{T}\alpha_{t}\beta_{t}h_{t}\right)\\ y=\arg\max(H_{\mathrm{final}}(x)) \end{array}$$

As shown in Figure 8, the area outlined by the dotted line, including noise reduction, feature extraction, classification, and attention weight calculation, is merged into the base classifier of the AdaBoost algorithm. Finally, *T* base classifiers are combined to form the final classifier. The final classifier was named the Denoising Fault Diagnosis Attention Network (DFAN).

## 4. Experimental Analysis

In this section, we describe the experiments performed on two different datasets to verify the noise robustness, generalization ability, and component effectiveness of the proposed DFAN. The DFAN network is based on Python3.7, Pytorch 1.10.0, CUDA 11.6, and NVIDIA GeForce RTX 3080.

### 4.1. Experiments Using Different Eccentricity Motors

In practice, it was found that the states of the motors on different pieces of artillery did not agree. We found that, after a certain period of use, the motors on the artillery have different degrees of eccentricity. Although the function of the motor does not affect use within a reasonable range of eccentricity, it still interferes with fault diagnosis.

In Experiment 1, several common motor failure states were realized by motors with different degrees of eccentricity. Experimental data were used to verify the effectiveness of the proposed method and explore the hyperparameters of the method.

#### 4.1.1. Description of the Experimental Environment and Data

The vibrational signals were collected for four different eccentricity states, namely, normal and 5%, 10%, and 15% eccentricity, at a sampling frequency of 10 kHz. Motor health status includes normal state, interturn short circuit, open-circuit fault, and bearing failure. In order to simulate the working state of the artillery mount, signals were collected under three loading conditions, corresponding to the load of the mount when the mount was unloaded, when the propellant was loaded, and when the standard bottom concave was loaded.

In each crossover condition, 600 samples were collected, 80% of which were used as the training set. Thus, there were 28,800 samples in the training set and 7200 samples in the test set under all cross-conditions. Each sample had 1024 points. The experimental setup is shown in Figure 9.

#### 4.1.2. Experimental Analysis

In the samples presented in Section 4.1.1, noise with SNRs of −4 dB, 0 dB, and 4 dB was further added. In this section, we describe how the DFAN method could better find the appropriate wavelet basis and hyperparameters without setting mixing noise that is inconvenient to quantify.

Eight fault diagnosis networks were constructed for comparison.

Based on ref. [16], we constructed a PCA–SVM as the most basic FFD model, and to be the baseline.

In a simplified model, the DWCN module in the DFAN method was substituted with a conventional convolutional layer (CNN). In an anti-noise model, the first layer of the previous model was replaced with a wide kernel convolution layer (WCNN). The classification networks of CNN and WCNN were constructed according to the method reported by ref. [29].

Using the method described in ref. [26], three mononuclear wavelet kernel networks with Morlet, Mexhat, and Laplacian wavelets as the kernels were constructed (named wavelet nets-M, wavelet nets-H, and wavelet nets-L, respectively).

Following the method in ref. [39] and based on transfer learning, we constructed a PCA–SVM and TrAdaBoost method.

In the DFAN method reported in this paper, the original channel *C*_0_ was set to 64, and the fused channel *C* was set to 32. In the fusion wavelet kernel, the Morlet wavelet occupied 22 primitive channels, and the Mexhat wavelet and Laplace wavelet occupied 20 primitive channels each. Their scaling parameters were initially set to be evenly distributed among different channels, with the Morlet, Mexhat, and Laplace values being [0.1, 3], [0.1, 4.5], and [0.1, 2], respectively.

The experiment was performed five times on each model to remove randomness.

In addition to comparing the classification accuracy of each model, the study introduced NI, which represents the robustness of the model against noise, defined as follows:(29)$$\mathrm{NI}=\frac{\Delta \mathrm{Acc}}{\Delta \mathrm{SNR}\cdot {\mathrm{Acc}}_{\mathrm{best}}}\left(\%\right)$$

Δ represents the difference between the best and worst accuracy; NI is inversely proportional to model robustness.

Figure 10 shows the classification performance of the different models. Detailed values are shown in Table 1.

As a very basic FFD model, PCA–SVM performed poorly under different noise conditions. Although PCA was also used in this method to align the samples before introducing the attention mechanism, the poor performance of the PCA–SVM model demonstrates that its performance had little to do with the introduction of the PCA steps. The TrAdaBoost method had similar NI values to the PCA–SVM model. However, since there were sufficient training samples, as an ensemble learning model, the classification accuracy was higher.

The main differences between Wavelet-M, Wavelet-H, Wavelet-L, CNN, WCNN, and the methods presented here, were the difference in signal processing in the first layer and the absence of attention-based AdaBoost ensemble learning in the subsequent layers. As an improved version of the CNN with noise reduction capability, the WCNN performed slightly better than the CNN under the three SNR conditions, which was in line with expectations.

Wavelet-M performed better than the other wavelet basis but more weakly than the DFAN. The proposed method achieved the best performance. This result confirms that although the effect of individual wavelet basis was not necessarily stronger than that of the WCNN, the combination of different wavelet basis improved the performance of the model, since the fused wavelet basis allocated more channels to the appropriate wavelet basis. The attention-mechanism-based AdaBoost method retained more possible wavelet basis for fault diagnosis and assigned higher weights for use. At the same time, the generalization and robustness of the model were also good. The fault accuracy was 92.3%, 95.9%, and 98.1% for the −4 dB, 0 dB, and 4 dB signals, respectively.

The settings of some DFAN hyperparameters are shown in Figure 11.

The appropriate number of epochs facilitates the fusion of wavelet kernels to update the weights of each basic wavelet basis, and 32 was also empirically consistent.

As the number of fused convolution kernels increased, the classification accuracy also slowly increased, indicating that this method is effective for some extreme samples. However, given the computational cost, setting the number at 32 was a more appropriate choice.

When there was only one base classifier, the proposed method degenerated into a WCNN-like model. When the number of base classifiers was increased, the accuracy rate increased rapidly, achieving its highest value at 7, which was 10% higher than when there was only one base classifier.

### 4.2. Experiments with an Automated Loading Test System

#### 4.2.1. Description of the Experimental Environment and Data

The motor-driven artillery automatic ammunition loading test system is shown in Figure 12. It included the host computer, PLC controller (PLC-X20 IF 1072 and PLC-X20 CP 0484, B&R, Eggelsberg, Austria), driver (SOL-GUIA50/100E, Elmo, Tokyo, Japan) and PMSM (ASM-V-05D-32, Siemens, Yangzhou, China). Metallurgical Automation Design and Research Institute servo Institute), and a reducer (reduction ratio of 150). The current loop sampling frequency was 1 kHz.

The data were divided into six categories, and the loads of the coordinated actions were all filled with standard module charges. Detailed data information is given in Table 2. Each test was recorded from the time the control signal was sent, and 1024 points were taken as samples to cover a coordinated action process. In model building and testing, 80% of the samples are the training set and 20% are the test set, resulting in 480 training samples and 120 test samples.

#### 4.2.2. Experimental Analysis

Noise was added to the samples in Section 4.2.1 below, and the SNR was set from −4 dB to 4 dB with an interval of 2 dB. After obtaining the expanded noisy sample set, resampling was performed on the noisy sample set to select six training sample sets.

The number of samples in each sample set was 1000, and the noise included was −4 dB, −2 dB, 0 dB, 2 dB, 4 dB, and mixed noise, respectively. The distribution of various fault conditions in the sample set was also balanced. The nine fault diagnosis networks were the same as those described in Section 4.1.2 for comparison.

The model performance at different SNRs is shown in Figure 13.

The second experiment was closer to actual gun current data, and almost all models performed poorly on the mixed-noise treatment.

The performances of the Wavelet-H and Wavelet-M models were almost the same. This is, in fact, because they focused on different fault types and the distribution of the two types of faults in the sample was approximately the same.

The classification performance of the CNN improved rapidly with increasing SNR, which indicates that robustness to noise was the main factor affecting its performance.

The Wavelet-L curve was similar to that of the CNN and illustrates the oxygen content of the noisy signal in Experiment 2 with the wavelet basis mismatched under low noise performance and CNN conditions.

The TrAdaBoost curve trend was similar to that of the CNN, but its overall performance was better than the CNN; this demonstrates that ensemble learning played a role in the study, but to achieve better results, the base classifier itself requires a certain degree of accuracy.

The DFN method excluded the attention mechanism and ensemble learning. In addition to the case of small noise, it was second only to the DFAN, which shows that it had the ability to denoise samples with mixed noise, but requires ensemble learning to further strengthen its performance.

The performance of WCNN in the mixed noise environment was second only to the DFN and DFAN, and in other cases, it performed similarly to the Wavelet-H and Wavelet-M kernels. Their main difference was also in their noise reduction ability.

The DFAN method still had obvious advantages in the mixed noise link, exhibiting the best classification performance in each case.

## 5. Conclusions

In terms of research into artillery motor fault diagnosis in complex environments, the proposed DFAN method demonstrated its ability to extract fault features from motor current data containing compound noise and classification faults. As a motor fault diagnosis method integrating the attention mechanism, AdaBoost ensemble learning, and wavelet fusion noise reduction, the DFAN method demonstrated better robustness to noise at all levels of SNR compared with conventional methods.

Each base classifier of the DFAN is trained through the global sample space, while the attention mechanism finds the most suitable base classifier for the current sample during the diagnosis of a particular sample. The classification performance of the DFAN in Experiment 1 was better than that in Experiment 2, as the AdaBoost method was able to improve the generalization of the classifier and lower the upper bound of the error rate by expanding the training sample size.

Particularly in the context of fault diagnosis of sample signals under mixed SNR conditions, DFAN’s comprehensive accuracy surpassed that of the second-best DFN method by 8.5%. The overall test results demonstrate a 14.6% higher comprehensive diagnostic accuracy compared with the mature TrAdaBoost method without a noise reduction module, highlighting the crucial role played by the noise reduction module and attention ensemble learning component in this model.

Although the DFAN method achieves the expected accuracy, the computational complexity of integrating wavelet fusion, attention mechanism, and AdaBoost ensemble learning is relatively high, leading to poor timeliness in fault diagnosis. In future research endeavors, it is imperative to explore simplified operational mechanisms while maintaining accuracy levels to enhance the timeliness of fault diagnosis.

## Figures and Tables

**Figure 1 sensors-24-00847-f001:**
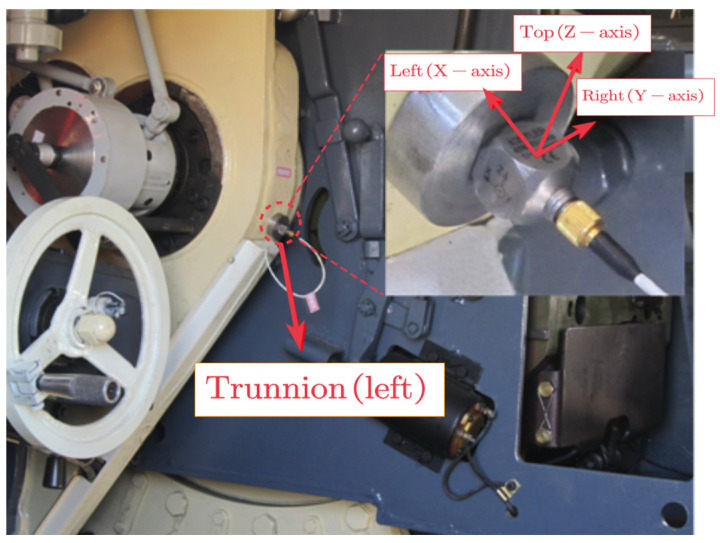
A vibration test in an artillery trunnion.

**Figure 2 sensors-24-00847-f002:**
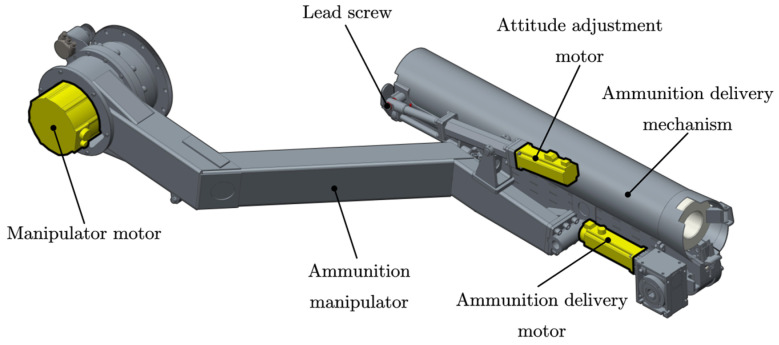
An automatic ammunition loading system.

**Figure 3 sensors-24-00847-f003:**
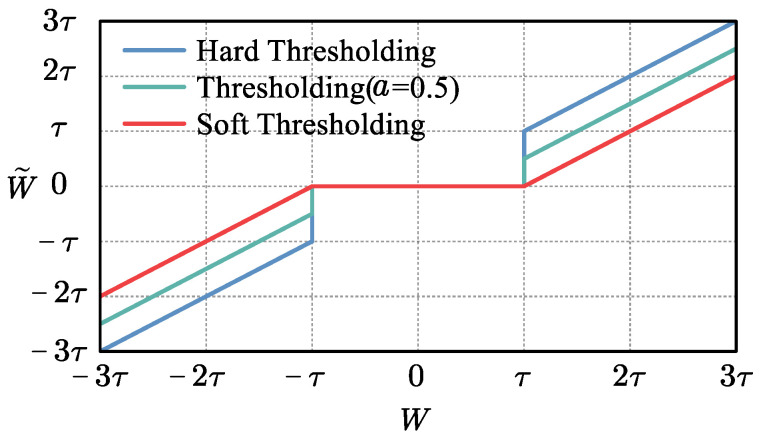
Wavelet threshold function.

**Figure 4 sensors-24-00847-f004:**
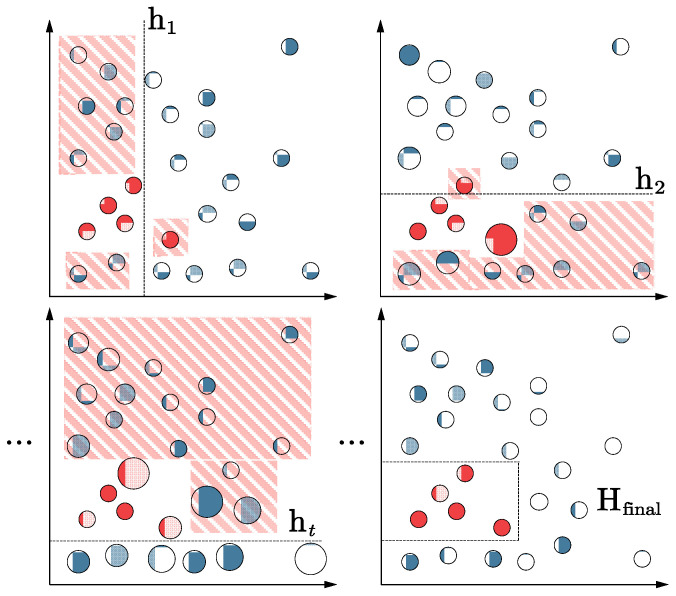
Example of the weighting procedure given to misclassified samples by the AdaBoost algorithm.

**Figure 5 sensors-24-00847-f005:**
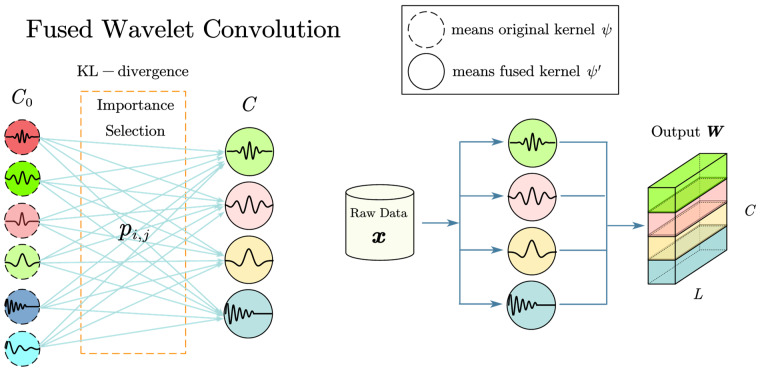
Fusion wavelet convolution generates a new wavelet kernel.

**Figure 6 sensors-24-00847-f006:**
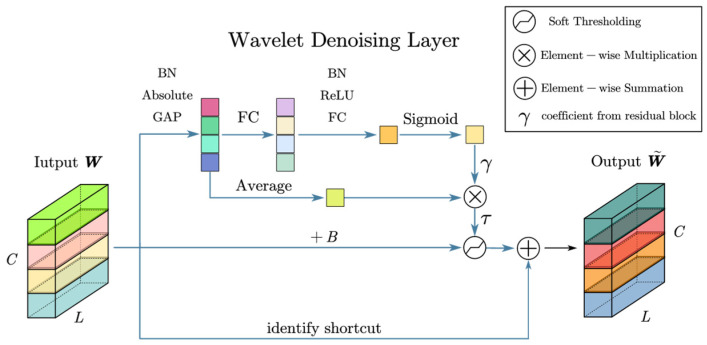
Residual shrinkage network for wavelet soft thresholding.

**Figure 7 sensors-24-00847-f007:**
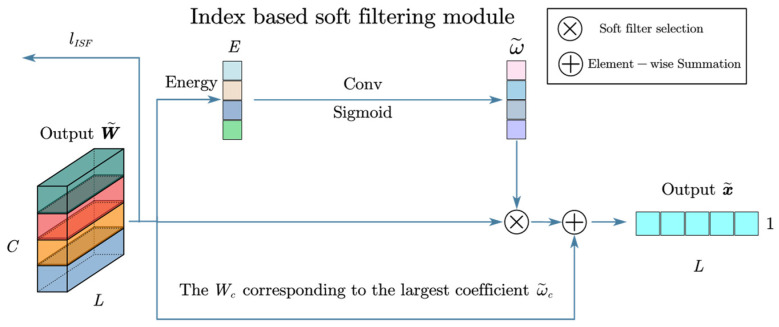
Index-based soft filtering module.

**Figure 8 sensors-24-00847-f008:**
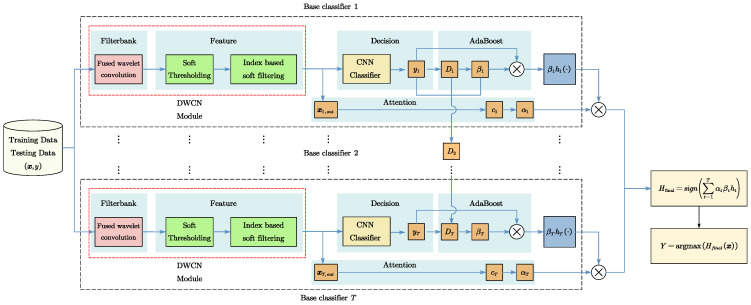
Flowchart of the fault diagnosis algorithm.

**Figure 9 sensors-24-00847-f009:**
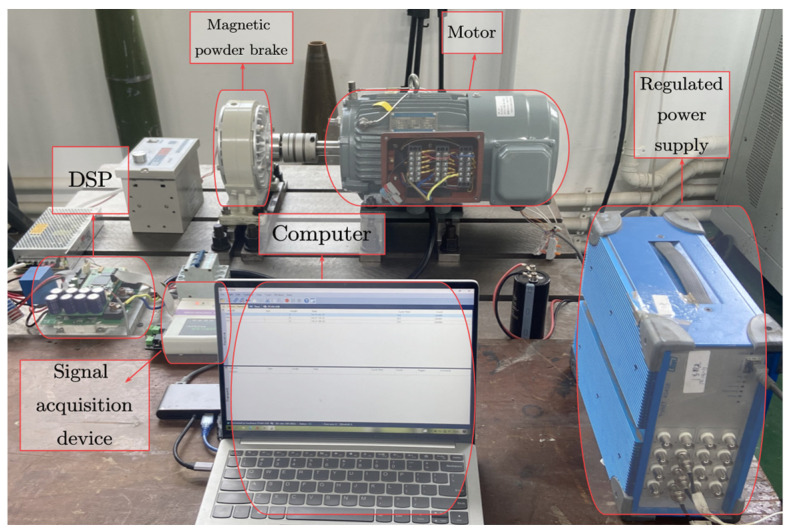
Fault diagnosis platform for an eccentric motor.

**Figure 10 sensors-24-00847-f010:**
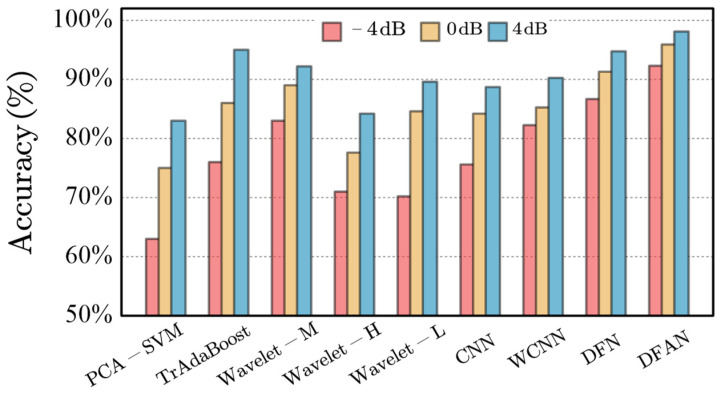
Classification performance of different models.

**Figure 11 sensors-24-00847-f011:**
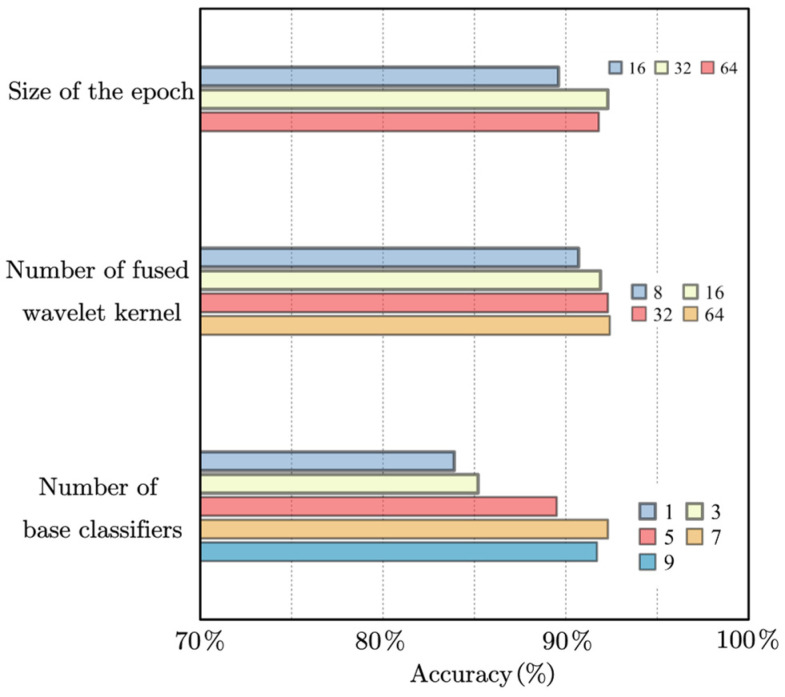
Effects of hyperparameters on model accuracy.

**Figure 12 sensors-24-00847-f012:**
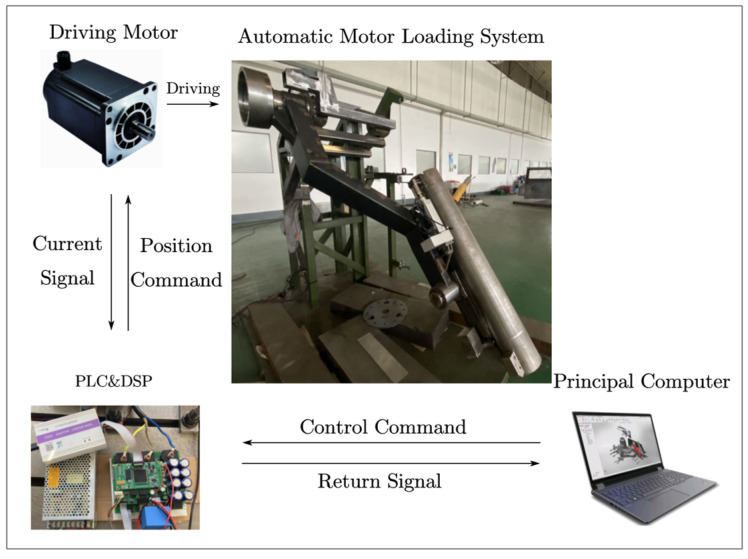
Automatic ammunition loading test system for artillery.

**Figure 13 sensors-24-00847-f013:**
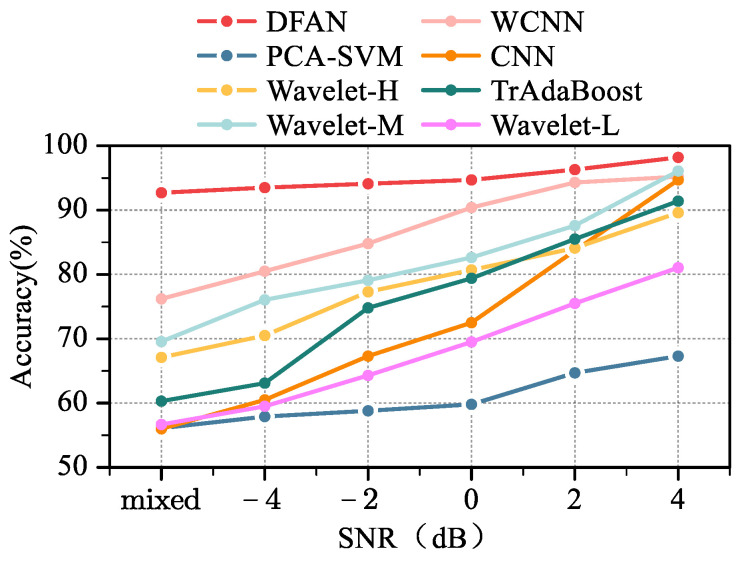
Model performance with different SNRs.

**Table 1 sensors-24-00847-t001:** Performance of classification models under different SNRS.

Model	Average Accuracy −4 dB	Average Accuracy 0 dB	Average Accuracy 4 dB	Noise Influence (NI)
PCA-SVM	63.36	75.67	83.05	2.96
TrAdaBoost	76.78	86.33	95.52	2.45
Wav-M	83.98	89.78	92.59	1.16
Wav-H	71.54	77.4	85.56	2.39
Wav-L	70.2	84.6	89.38	2.68
CNN	75.68	84.2	88.1	1.44
WCNN	82.25	85.15	90.95	1.20
DFN	86.67	91.3	94.74	1.06
DFAN	92.3	95.9	98.1	0.74

**Table 2 sensors-24-00847-t002:** Distribution of the number of samples in each class.

Barrel Angle	Fault Condition	Sample Size
0°	Motor inter-turn short circuit	80
36°	Motor inter-turn short circuit	80
0°	Broken teeth of reduction gear	70
36°	Broken teeth of reduction gear	70
0°	Normal state	150
36°	Normal state	150

## Data Availability

No new data were created or analyzed in this study. Data sharing is not applicable to this article.

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
