# Peer review of "Fault Diagnosis Methods for an Artillery Loading System Driving Motor in Complex Noisy Environments"

_sensors, 2024, doi:10.3390/s24030847_

Round 1

Reviewer 1 Report

Comments and Suggestions for Authors

1. Avoid using the subjective expressions. 'We' should be replaced by using 'this study'.

2. Keywords missing 'noise'

3. Literature review on DL based motor fault diagnosis far from  enough. Seem only Chinese researchers are considered.

4. Authors should let readers know that the DFAN was proposed at the beginning of the manuscript.

5. Figure 8, what are the base classifier used in this study?

6. Table 1, why used PCA-SVM, WCNN for comparison? Why not compare DFAN with DFN (DFAN without attention)? So readers can understand  the detailed effects of each part in  the proposed DFAN method. Besides, the role of ensemble learning is not clear.

7. Table 2, why only barrel angles of 0 and 36?Authors should mentioned that whether the impact of imbalanced sample size for various fault classes should be considered or not.

8. The Conclusions should be revised by adding more numerical findings.

Comments on the Quality of English Language

A language edition should be suggested. Some of the sentences are too long and hard to be understood.

Author Response

The author's replies are written in a Word document. 

Reviewer 2 Report

Comments and Suggestions for Authors

In this paper, a fault diagnosis model based on attention mechanism, AdaBoost method, and wavelet noise reduction network is proposed. The model is used to address the difficulty of obtaining high-quality motor signals in complex noisy interference environments. The proposed method achieves an average accuracy of 92 percent, more than 9 percent higher than the conventional method, and the proposed method is proven to have better generalization and reliability.

The main part of the paper is useful for specialists in fault diagnosis. But the paper is written very carelessly that makes it very difficult for reading. The authors use the arrangements AdaBoost and adaboost, basis and bases (line 305), the sentence on line 314 is impossible to understand, and so on.

Some other remarks.

1.      (1) and (2) should be explained or referenced.

2.      W with ~ is absent in (6).

3.      AdaBoost should be referenced.

4.      What is a problem mentioned just before (12)?

5.      What is k in (26)?

6.      English should be improved significantly.

The paper can be accepted only after significant improvement.

Comments on the Quality of English Language

No

Author Response

(The authors gave the same response as above.)

Round 2

Reviewer 1 Report

Comments and Suggestions for Authors

Actually, most of the comments have been revised. But the English presentation I am not sure. This is the only comment I think the manuscript should be improved.

Reviewer 2 Report

Comments and Suggestions for Authors

No

Comments on the Quality of English Language

No